# Identification of novel genetic susceptibility loci for thoracic and abdominal aortic aneurysms via genome-wide association study using the UK Biobank Cohort

Tamara Ashvetiya◉, Sherry X. Fan◉, Yi-Ju Chen, Charles H. Williams, Jeffery R. O'Connell, James A. Perry◉*, Charles C. Hong◉*

Department of Medicine, University of Maryland School of Medicine, Baltimore, Maryland, United States of America

◉ These authors contributed equally to this work.
* jperry@som.umaryland.edu (JAP); charles.hong@som.umaryland.edu (CCH)

**Data Availability Statement:** All relevant data are within the paper and its Supporting information files.

## Abstract

### Background

Thoracic aortic aneurysm (TAA) and abdominal aortic aneurysm (AAA) are known to have a strong genetic component.

### Methods and results

In a genome-wide association study (GWAS) using the UK Biobank, we analyzed the genomes of 1,363 individuals with AAA compared to 27,260 age, ancestry, and sex-matched controls (1:20 case:control study design). A similar analysis was repeated for 435 individuals with TAA compared to 8,700 controls. Polymorphism with minor allele frequency (MAF) >0.5% were evaluated.

We identified novel loci near *LINC01021*, *ATOH8* and *JAK2* genes that achieved genome-wide significance for AAA (p-value <5x10⁻⁸), in addition to three known loci. For TAA, three novel loci in *CTNNA3*, *FRMD6* and *MBP* achieved genome-wide significance. There was no overlap in the genes associated with AAAs and TAAs. Additionally, we identified a linkage group of high-frequency variants (MAFs ~10%) encompassing *FBN1*, the causal gene for Marfan syndrome, which was associated with TAA. In FinnGen PheWeb, this *FBN1* haplotype was associated with aortic dissection. Finally, we found that baseline bradycardia was associated with TAA, but not AAA.

### Conclusions

Our GWAS found that AAA and TAA were associated with distinct sets of genes, suggesting distinct underlying genetic architecture. We also found association between baseline brady-cardia and TAA. These findings, including *JAK2* association, offer plausible mechanistic and therapeutic insights. We also found a common *FBN1* linkage group that is associated

**Funding:** This work was supported by National Institute of General Medical Sciences, R01GM118557 and National Heart, Lung, and Blood Institute, R01HL1351291 to CCH, and National Heart, Lung, and Blood Institute, 1U01HL137181 to JP. The funders had no role in study design, data collection and analysis, decision to publish, or preparation of the manuscript.

**Competing interests:** The authors have declared that no competing interests exist.

**Abbreviations:** AA, aortic aneurysm; AAA, abdominal aortic aneurysm; GWAS, genome-wide association study; ICD, international classification of diseases; LD, linkage disequilibrium; MAF, minor allele frequency; MVP, Million Veteran Program; PC, principal component; SNP, single nucleotide polymorphism; TAA, thoracic aortic aneurysm; TAAD, thoracic aortic aneurysms and dissection; UK, United Kingdom.

with TAA and aortic dissection in patients who do not have Marfan syndrome. These *FBN1* variants suggest shared pathophysiology between Marfan disease and sporadic TAA.

## Introduction

Aortic aneurysms (AA) carry a significant burden of morbidity and mortality. In 2018, thoracic aortic aneurysms (TAA) and abdominal aortic aneurysms (AAA) together were responsible for 9,923 deaths in the United States, typically from complications such as aortic dissection and rupture [1]. AA primarily affect elderly males in the sixth and seventh decades of life with risk factors of tobacco use, hypertension and atherosclerosis [1]. However, AAs found in patients younger than 65 years are more often attributed to genetic predisposition. For AAA, individuals with first-degree family members with the disease are at two-fold risk of developing AAA as compared to patients with no family history [2]. Genetic studies of TAA and AAA have revealed genetic heterogeneity and polygenic inheritance patterns with variable disease penetrance [2–4].

For thoracic aortic aneurysms and dissection (TAAD), rare monogenetic syndromic disorders such as Marfan, Ehlers-Danlos and Loeys-Dietz syndromes are known to dramatically increase risk in younger individuals [5]. This risk results from mutations in *FBN1*, *COL3A1*, and genes that encode TGF- β signaling proteins respectively [5]. Beyond these, up to one-fifth of patients with TAAD have a familial predisposition toward aneurysmal disease [5]. For instance, mutations in *FBN1* (fibrillin-1), the causal gene for Marfan syndrome, may increase the risk of thoracic aortic aneurysms or dissections even in individuals who do not have Marfan syndrome [4,6,7]. Additionally, rare mutations in *ACTA2*, encoding smooth muscle protein alpha ($\alpha$)-2 actin, may account for up to 14% of familial forms of TAAs [8]. Furthermore, genetic studies of TAAD show it is closely associated with bicuspid aortic valve, another condition with strong heritability [5,9].

Since AA has a strong genetic component in certain individuals, an enhanced understanding of these factors may ultimately aid the early detection of this silent disease before it progresses into life-threatening aortic dissections and ruptures. There is evidence to suggest that a strong personal or family history of aneurysms or dissection in individuals under the age of 50 should be an indication for genetic testing to diagnose inherited aortopathy [10]. In appropriately selected patients with suspected familial aneurysmal disease, the yield of genetic testing could be as high as 36% [10]. Yet, much of the underlying genetic risk factors remain unknown.

In this paper, we address the discovery of novel genetic loci that may portend increased risk for the development of thoracic and abdominal aortic aneurysms based on a genome-wide association study (GWAS) using data from the UK Biobank [11]. The UK Biobank allows for powerful association studies with tremendous potential for expanding knowledge on the genetic basis of aortic aneurysms, as well as uncovering novel genetic variants associated with clinical diseases.

## Methods

### Ethical approval

The present study, which involved deidentified data obtained from the UK Biobank Resource under Application Number 49852, received the proper ethical oversight, including the

determination by the University of Maryland, Baltimore Institutional Review Board that the study is not human research (IRB #: HF-00088022).

## Study population

The UK Biobank, which was used for the GWAS presented here, is a large, ongoing prospective cohort study that recruited 502,682 UK participants between 2006–2010, ranging in age from 40–69 years at the time of recruitment. Extensive health-related records were collected from these participants, including clinical and genetic data, with over 820,000 genotyped single nucleotide polymorphisms (SNPs) and up to 90 million imputed variants available for most individuals. We carried out a genome-wide association study using the UK Biobank to interrogate the genome for statistically significant associations between SNPs and clinical manifestations of abdominal and thoracic aortic aneurysms at the population level.

## Genome-wide association study (GWAS)

Using data from the UK Biobank Resource on 487,310 subjects with imputed genotypes, we performed quality control by removing those with genetic relatedness exclusions (Data-Field 22018—UKB, https://biobank.ctsu.ox.ac.uk/crystal/field.cgi?id=22018; 1532 subjects), sex chromosome aneuploidy (Data-Field 22019 –UKB, https://biobank.ctsu.ox.ac.uk/crystal/field.cgi?id=22019; 651 subjects), mismatch between self-reported sex and genetically determined sex (Data-Field 31 –UKB, https://biobank.ctsu.ox.ac.uk/crystal/field.cgi?id=31; Data-Field 22001 –UKB, https://biobank.ctsu.ox.ac.uk/crystal/field.cgi?id=22001; 372 subjects), recommended genomic analysis exclusions (Data-Field 22010—UKB, https://biobank.ctsu.ox.ac.uk/crystal/field.cgi?id=22010; 480 subjects), and outliers for heterozygosity or missing rate (Data-Field 22017 –UKB, https://biobank.ctsu.ox.ac.uk/crystal/field.cgi?id=22077; 968 subjects). For "cases" we selected subjects with the following ICD10 (international classification of diseases) diagnostic codes, classified as either "main" or "secondary": "abdominal aortic aneurysm, without mention of rupture" (1,363 patients, ICD10 code I71.4) and "abdominal aortic aneurysm, ruptured" (131 patients, ICD10 code I71.3). The selected set was purged of relatedness by removing one of each related pair in an iterative fashion until no related subjects remained. A pool of possible control subjects was generated by removing the cases and removing subjects with ICD10 codes listed as "excluded from controls" in S1 Table. Subjects with these ICD10 codes were removed from the population of controls to avoid introducing confounding factors, specifically the TAA and aortic valvular disorders, in the analysis. The pool was further reduced by removing related subjects. From the resulting reduced pool of possible controls, 20 control subjects were selected for each case subject, matched by sex, age and ancestry with sex as a required match (n = 27,260 controls). Incremental tolerances were used for age and ancestry with tolerances being expanded with each iteration until the desired number of matching controls were found for each case subject. The age tolerance ranged from 0 (exact match) up to 7 years. Ancestry matching was performed using principal components (PCs) supplied by the UK Biobank. The mathematical distance in a graph created by plotting the PC1 and PC2 eigenvalues provided by the UK Biobank was used to test ancestry similarity. The ancestry "distance" tolerated ranged from 2 PC units up to a maximum of 80 PC units (S1 Fig), where PC1 ranged from 0 to +400 and PC2 ranged from -300 to +100 units. Using these tolerances, 20 matching controls were found for every case. The analysis was repeated with patients who carried either a main or secondary diagnosis of "thoracic aortic aneurysm, without mention of rupture" (435 patients, ICD10 code I71.2) or "thoracic aortic aneurysm, ruptured" (22 patients, ICD10 code I71.1). Case to control ratio was again set at 1:20 (n = 8,700 controls), and patients with the diagnoses listed in S2 Table were excluded from the control population.

Subjects with these ICD10 codes were removed from the population of controls to avoid introducing confounding factors, specifically the AAA and aortic valvular disorders, in the analysis.

The association analysis was performed with PLINK2 using logistic regression [12]. The thoracic aortic aneurysm (TAA) and abdominal aortic aneurysm (AAA) phenotypes were run against 40 million imputed variants supplied by the UK Biobank with imputation quality scores greater than 0.70. The analysis included covariates of sex, age, and principal components 1 through 5 to adjust for ancestry. Pre-calculated PC data for the first 40 principal components was supplied by the UK Biobank. Our preliminary analysis showed that only the first 5 PCs had significance with p-values less than 0.05. Thus, we used only the first 5 PCs in our GWAS.

### Identification of significant SNPs for AAA and TAA Phenotypes

The SNPs identified in the analysis were filtered to include only those with minor allele frequency (MAF) of at least 0.5% and p-value $<1 \times 10^{-6}$, which is suggestive of genome-wide significance.

## Results

### Abdominal aortic aneurysm

Of the 1,363 individuals documented to have abdominal aortic aneurysms, 131 (9.61%) had rupture of the aneurysm (Table 1). Their baseline characteristics versus matched controls are shown in Table 1. The affected patients ranged in age at diagnosis from 42.42 to 79.04 years (mean = 68.08); 86.72% were male and 13.28% female. Genetic ancestry was predominantly British (93.54%); however, patients were also represented from Irish (2.64%), Indian (0.22%), Caribbean (0.51%), and African (0.15%) backgrounds.

Based on GWAS of these 1,363 individuals, we identified four independent loci near the *LINC01021*, *ADAMTS8*, *ATOH8* and *JAK2* genes (7 SNPs in total) that achieved a genome-wide significance level of p-value $<5 \times 10^{-8}$ for variants with MAF $\geq$0.5% (Fig 1A). Of these, *LINC01021*, *ADAMTS8* and *JAK2* represent novel loci associated with AAA, and each of the four loci harbors SNPs that possess features suggestive of functional importance, with

**Table 1. Basic demographics and characteristics of individuals with abdominal aortic aneurysm (AAA) and thoracic aortic aneurysm (TAA), and the respective matched (for age, sex, ancestry) controls in UK Biobank.**

| | AAA Cases | Matched* Controls | TAA Cases | Matched* Controls |
|---|---|---|---|---|
| **Individuals (#)** | 1363 | 27260 | 435 | 8700 |
| **# with rupture** | 131 (9.6%) | 0 (0%) | 22 (5.1%) | 0 (0%) |
| **Mean Age (yr) at diagnosis or assessment*** | 68.1 | 68.1 | 65.3 | 65.3 |
| **Male (%)*** | 86.7 | 86.7 | 68.7 | 68.7 |
| **British Ancestry (%)*** | 93.5 | 93.5 | 87.6 | 87.6 |
| **BMI** | 28.7** | 27.8 | 28.0 | 27.7 |
| **Height (in)** | 68.2 | 68.1 | 67.9** | 67.3 |
| **Waist (in)** | 39.38*8 | 37.9 | 37.6 | 36.9 |
| **Weight (lbs)** | 191** | 184 | 184 | 179 |
| **Diastolic BP** | 83.3 | 83.3 | 81.3** | 83 |
| **Systolic BP** | 145 | 147 | 143 | 144 |
| **Pulse Rate** | 69.8 | 69 | 67** | 69.2 |

*Characteristics that were matched.

**P-value <0.05 in comparison to matched controls.

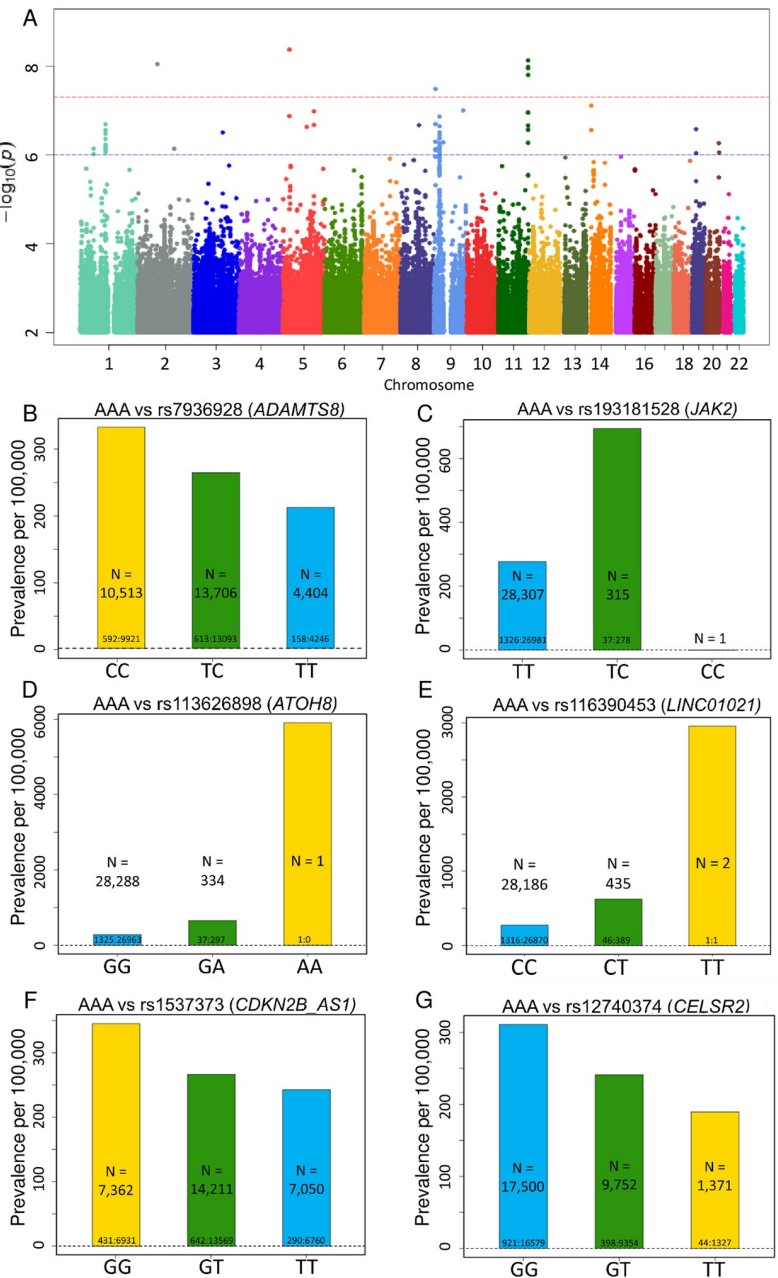

**Fig 1. Top SNPs associated with AAA. (A)** Manhattan plot of GWAS results (MAF >0.5%) for AAA. Significance is displayed on the y-axis as -log$_{10}$ of the p-value, with results ordered along the x-axis by chromosome (each bar represents a different chromosome). **(B-G)** Prevalence of abdominal aortic aneurism (AAA) per 100,000 participants in the UK Biobank by genotype. Bars labeled with ratio of cases: Controls. **(B)** Prevalence of AAA decreases with *ADAMTS8* variant rs7936928 status (P-value = 7.51x10$^{-9}$, OR per T allele = 0.786). Decrease in AAA prevalence is noted in the homozygotes for the minor allele (T/T) in comparison to the heterozygotes (C/T) and the noncarriers (C/C) in a stepwise, "dosage-dependent" manner. **(C)** Prevalence of AAA increases with *JAK2* variant rs193181528 status (P-value = 3.26x10$^{-8}$, OR per C allele = 2.776). **(D)** Prevalence of AAA increases with *ATOH8* variant rs113626898 status (P-value = 9.06x10$^{-9}$, OR per A allele = 2.714). **(E)** Prevalence of AAA increases with *LINC01021* variant rs116390453 status (P-value = 4.26 x10$^{-9}$, OR per T allele = 2.505). **(F)** Prevalence of AAA decreases with *CDKN2B-AS1* variant rs1537373 status (P-value = 6.68x10$^{-7}$, OR per T allele = 0.8211). **(G)** Prevalence of AAA decreases with *CELSR2* variant rs12740374 status (P-value = 2.04x10$^{-7}$, OR per T allele = 0.7668).

**Table 2. Top SNPs associated with abdominal aortic aneurysm.**

| SNP | Chr: BP (GRCh37) | Allele | Nearest Gene | Type | MAF (%) | OR (95% CI) | P-value |
|---|---|---|---|---|---|---|---|
| rs116390453 | 5:27,997,008 | C/T | *LINC01021* | Intergenic | 0.77 | 2.505 (1.84–3.4) | 4.26x10⁻⁹ |
| rs7936928 | 11:130,279,168 | C/T | *ADAMTS8* | Intronic | 39.3 | 0.786 (0.724–0.853) | 7.51x10⁻⁹ |
| rs4936099 | 11:130,280,725 | A/C | *ADAMTS8* | Intronic | 40.7 | 0.789 (0.727–0.856) | 1.05x10⁻⁸ |
| rs11222084 | 11:130,273,230 | A/T | *ADAMTS8* | Intergenic | 36.7 | 0.785 (0.723–0.853) | 1.12x10⁻⁸ |
| rs3740888 | 11:130,278,210 | T/C | *ADAMTS8* | Intronic | 39.4 | 0.790 (0.728–0.858) | 1.59x10⁻⁸ |
| rs113626898 | 2:86,015,431 | G/A | *ATOH8* | UTR3 | 0.59 | 2.714 (1.93–3.82) | 9.06x10⁻⁹ |
| rs193181528 | 9:5,059,543 | T/C | *JAK2* | Intronic | 0.58 | 2.776 (1.93–3.99) | 3.26x10⁻⁸ |
| rs1537373 | 9:22,103,341 | G/T | *CDKN2B-AS1* | ncRNA | 49.4 | 0.821 (0.76–0.887) | 6.68x10⁻⁷ |
| rs12740374 | 1:109,817,590 | G/T | *CELSR2* | 3'-UTR | 21.8 | 0.767 (0.694–0.848) | 2.04x10⁻⁷ |
| rs629301 | 1:109,818,306 | T/G | *CELSR2* | 3'-UTR | 22 | 0.770 (0.697–0.851) | 2.78x10⁻⁷ |
| rs646776 | 1:109,818,530 | T/C | *CELSR2* | downstream | 22 | 0.770 (0.697–0.851) | 2.83x10⁻⁷ |
| rs3832016 | 1:109,818,158 | C/CT | *CELSR2* | 3'-UTR | 21.3 | 0.768 (0.694–0.85) | 3.35x10⁻⁷ |
| rs660240 | 1:109,817,838 | C/T | *CELSR2* | 3'-UTR | 21.3 | 0.771 (0.696–0.853) | 4.36x10⁻⁷ |
| rs7528419 | 1:109,817,192 | A/G | *CELSR2* | 3'-UTR | 22 | 0.777 (0.703–0.858) | 6.81x10⁻⁷ |

7 variants in 4 genes, *LINC01021*, *ADAMTS8*, *ATOH8* and *JAK2*, reached genome-wide significance P-value of $< 5 \times 10^{-8}$, while 7 additional variants in *CDKN2B-AS1* and *CELSR2*, while not statistically significant, replicated findings from earlier studies. Of these, *LINC01021*, *ATOH8* and *JAK2* are novel AAA-associated loci identified in the present study (bold faced). Chr:BP denotes the chromosome location and NCBI Build 37 SNP physical position. Variants that are in linkage disequilibrium (LD) are identically colored. MAF, minor allele frequency. OR, odds ratio.

biological plausibility as disease susceptibility loci (Table 2). In addition, we found 7 additional variants in *CDKN2B-AS1* and *CELSR2*, while not reaching genome-wide significance, are within the suggestive threshold for significance (p-value $<1 \times 10^{-6}$), replicate findings from earlier studies and possess a strong basis for biologic plausibility (Table 2). Full GWAS results for AAA are included in S5 Table for variants with MAF $\geq 0.5\%$ and p-value $<1 \times 10^{-6}$. Quantile-quantile plots (QQ Plots) are provided in S2 Fig to illustrate that the GWAS quality was well controlled.

Among the SNPs with genome-wide significance, we identified a linkage group of 4 variants in close proximity to the *ADAMTS8* gene, which encodes the ADAM metallopeptidase with thrombospondin type 1 motif 8, an inflammation-regulated enzyme expressed in macrophage-rich areas of atherosclerotic plaques [13] (Table 2 and Fig 1B; p-values $7.51 \times 10^{-9}$–$1.59 \times 10^{-8}$, MAF 36.7%-40.7%, and odds ratios 0.785–0.790). Prior studies have described upregulation of *ADAMTS8* in the macrophages of patients with abdominal aortic aneurysms [14]. Of note, the *ADAMTS8* locus was recently identified among 14 novel AAA-risk loci identified from a study of the Million Veteran Program [15].

Other notable variants identified in this genome-wide analysis include the intronic variant rs193181528 (Table 2 and Fig 1C; p-value $3.26 \times 10^{-8}$, MAF 0.58%, OR 2.776), located within the gene encoding JAK2 tyrosine kinase. Mutations in *JAK2* are suspected to play a potential role in the progression of AAAs [16,17]. Among human aortic tissues collected from patients undergoing AAA surgery, *JAK2* expression levels were higher in patients with AAA as compared to controls [16]. Treatment with JAK2/STAT3 pathway inhibitors attenuated experimental AAA progression by reducing the expression of pro-inflammatory cytokines and matrix metalloproteinases as well as inflammatory cell infiltration [16,17].

The analysis also identified variant rs113626898 (Table 2 and Fig 1D; p-value $9.06 \times 10^{-9}$, MAF 0.59%, OR 2.714) in the gene encoding atonal bHLH transcription factor 8 (*ATOH8*), which plays a role in myogenesis and contributes to endothelial cell differentiation,

proliferation and migration [18]. Dysregulation of this gene could plausibly contribute to aneurysmal formation given its important role in the cell cycles of myocytes and endothelial cells.

Variant rs116390453 (Table 2 and Fig 1E; p-value 4.26 x $10^{-9}$, MAF 0.77%, OR 2.505) is within a long intergenic non-coding RNA (*LINC01021*), also known as p53 upregulated regulator of p53 levels (PURPL). To date, PURPL has not been linked to aneurysmal formation.

In addition, we identified several distinct variants that do not reach the standard threshold for genome-wide significance for association with AAA, but are nevertheless within the suggestive threshold for genome-wide significance (p-value <1 x $10^{-6}$) (Table 2). Variant rs1537373 (Fig 1F; p-value 6.68 x $10^{-7}$, MAF 49.9%, OR 0.821) in *CDKN2B-AS1*, encoding the long non-coding RNA known as cyclin dependent kinase inhibitor CDKN2B antisense RNA1, is located within the *CDKN2B-CDKN2A* gene cluster at chromosome 9p21, a major genetic susceptibility locus for coronary artery disease, atherosclerosis and myocardial infarction [19]. This locus has also been previously associated with intracranial aneurysm and AAA formation [15,20–23]. Thus, *CDKN2B-AS1* variant rs1537373 may increase the risk of AAA formation indirectly through the development of atherosclerosis, a major clinical risk factor for AAA.

Finally, our analysis identified a linkage group of six SNPs within the *CELSR2* gene, encoding the cadherin EGF LAG seven-pass G-type receptor 2 (Table 2 and Fig 1G; p-values 2.04 x10-7–6.81 x10-7, MAF 21.3–22%, OR 0.767–0.777). While *CELSR2* SNPs do not meet traditional P-value cutoff for genome-wide significance, our findings corroborate prior GWAS associations of *CELSR2* with AAA [15,24,25], and identify new common SNPs that are associated with AAA (specifically rs3832016 and rs660240) (Table 2).

## Thoracic aortic aneurysm

Of the 435 individuals with thoracic aortic aneurysms, 22 (5.06%) had rupture of the aneurysm (Table 1). The affected patients ranged in age at diagnosis from 36.47 to 78.65 years (mean = 65.28); 68.74% were male and 31.26% female, with genetic ancestry of British (87.59%), Irish (2.07%), Indian (1.61%), Caribbean (1.38%), and African (0.23%) origins. Based on GWAS, we identified three SNPs that achieved a genome-wide significance level (p-value <5 x $10^{-8}$) together with a MAF ≥0.5% (Fig 2A and Table 3). Full GWAS results for TAA are included in S6 Table for variants with MAF ≥ 0.5%. Quantile-quantile plots (QQ Plots) are provided in S2 Fig to illustrate that the GWAS quality was well controlled.

Among the SNPs with significant p-values, variant rs149014140 in *CTNNA3* gene is of particular interest (Fig 2B; p-value 1.82 x $10^{-8}$, MAF 0.78%, OR 4.268) since *CTNNA3* encodes a vinculin/alpha-catenin family protein known to play a role in cell-to-cell adhesion of muscle cells [26]. Another significant variant is rs148927240 (Fig 2C; p-value 2.19 x $10^{-8}$, MAF 0.71%, OR 4.23), an intergenic variant located between the long non-coding RNA FERM domain containing 6 (*FRMD6*), involved in cell contact inhibition and cell cycle regulation [27], and the gene encoding one of the gamma subunits of a guanine nucleotide-binding protein (*GNG2*). Finally, variant rs78851735 (Fig 2D; p-value 3.79 x $10^{-8}$, MAF 0.98%, OR 3.446) is an intronic variant within the gene encoding myelin basic protein (*MBP*), which is the major protein in myelin sheaths of the nervous system [28]. The biological relevance of *FRMD6*, *GNG2* and *MBP* with respect to the development of aortic aneurysms is unclear at this time.

In addition, we identified a linkage group of high-frequency variants (MAFs 9.56–9.97%, odds ratios 1.615–1.644) that do not reach the standard threshold for genome-wide significance for association with TAA, but fall in within *FBN1, which encodes* the fibrillin-1 protein *FBN1* encodes the fibrillin-1 protein and is implicated in the pathogenesis of Marfan syndrome [29] (Table 4). Fibrillin-1 is important in maintaining the integrity of connective tissues

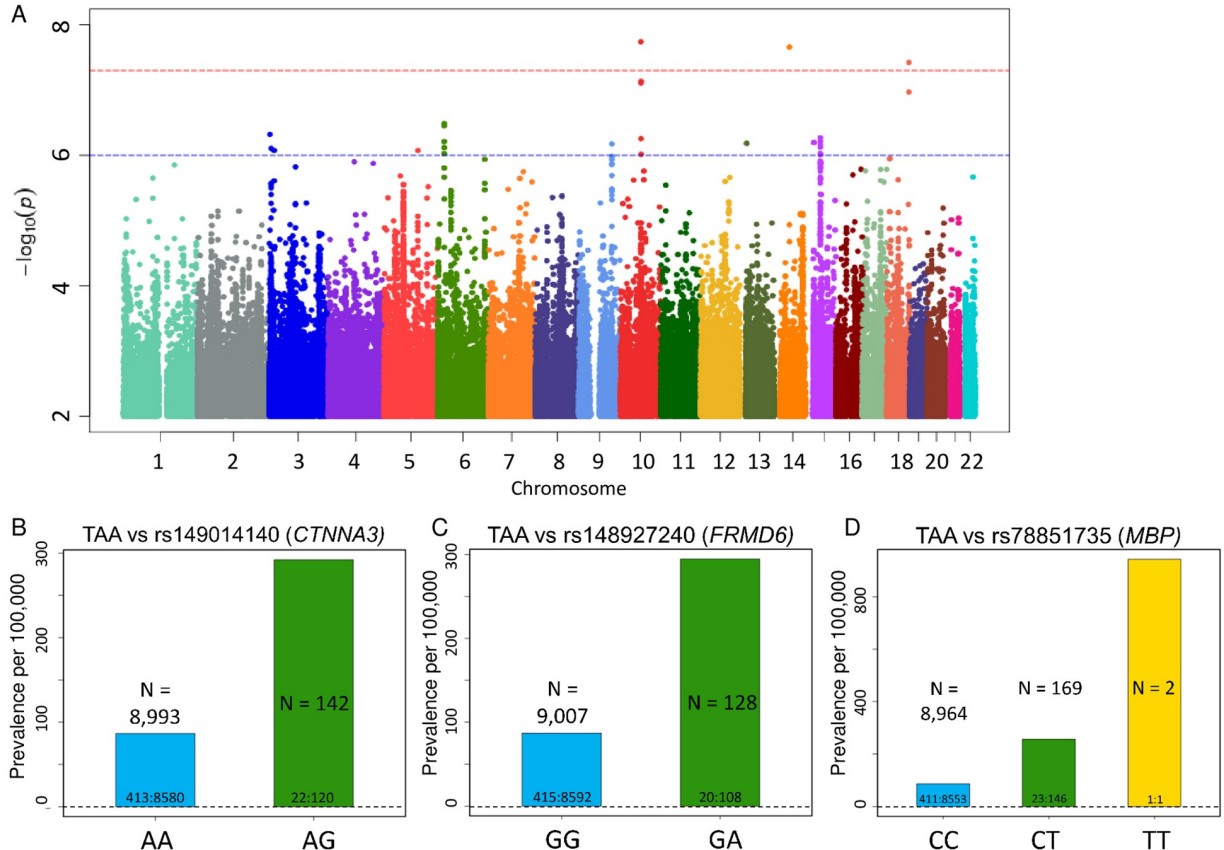

**Fig 2. Top SNPs associated with TAA. (A)** Manhattan plot of GWAS results (MAF >0.5%) for TAA. **(B-D)** Prevalence of thoracic aortic aneurism (AAA) per 100,000 participants in the UK Biobank by genotype. Bars labeled with ratio of cases: Controls. **(B)** Prevalence of TAA increases with *CTNNA3* variant rs149014140 status (P-value = 1.82x10⁻⁸, OR per G allele = 4.268). **(C)** Prevalence of TAA increases with *FRMD6* variant rs148927240 status (P-value = 2.19x10⁻⁸, OR per A allele = 4.23). **(D)** Prevalence of TAA increases with *MPB* variant rs78851735 status (P-value = 3.79x10⁻⁸, OR per T allele = 3.446).

throughout the body, as it serves as a structural component of calcium-binding myofibrils [29].

Interestingly, this haplotype, as illustrated by the *FBN1* intronic variant rs1561207, demonstrated a pronounced dose-dependence: homozygotes had significantly higher prevalence of thoracic aortic aneurism than heterozygotes (Fig 3A). Comorbid conditions of TAA patients with these *FBN1* variants (S3 Table), as well as gender and age at diagnosis (S4 Table) were similar to the overall population of TAA patients. In FinnGen PheWeb, comprised of genetic

**Table 3. SNPs in 3 novel loci associated with thoracic aortic aneurysm.**

| SNP | Chr:BP | Allele | Nearest Gene | Type | MAF (%) | OR (95% CI) | P-value |
|---|---|---|---|---|---|---|---|
| rs149014140 | 10:68,863,297 | A/G | *CTNNA3* | Intronic | 0.78 | 4.268 (2.57–7.07) | 1.82x10⁻⁸ |
| rs148927240 | 14:52,239,510 | G/A | *FRMD6* | Intergenic | 0.71 | 4.23 (2.55–7.01) | 2.19x10⁻⁸ |
| rs78851735 | 18:74,774,680 | C/T | *MBP* | Intronic | 0.98 | 3.446 (2.22–5.35) | 3.79x10⁻⁸ |

*CTNNA3*, *FRMD6* and *MBP* are novel TAA-associated loci identified in the present study (bold faced). Chr:BP denotes the chromosome location and NCBI Build 37 SNP physical position. MAF, minor allele frequency; OR, odds ratio.

**Table 4. Linkage group of *FBN1* variants associated with thoracic aortic aneurysm.**

| SNP | Chr: BP | Allele | Type | MAF (%) | OR (95% CI) | P-value |
|---|---|---|---|---|---|---|
| rs1561207 | 15: 48,858,971 | G/T | Intronic | 9.89 | 1.615 (1.33–1.96) | $1.57 \times 10^{-6}$ |
| rs689304 | 15: 48,922,360 | C/T | Intronic | 9.93 | 1.642 (1.35–1.99) | $5.38 \times 10^{-7}$ |
| rs625034 | 15: 48,926,202 | T/C | Intronic | 9.9 | 1.64 (1.35–1.99) | $6.15 \times 10^{-7}$ |
| rs1036476 | 15: 48,914,775 | T/C | Intronic | 9.89 | 1.636 (1.35–1.99) | $6.93 \times 10^{-7}$ |
| rs2028109 | 15: 48,919,103 | A/C | Intronic | 9.9 | 1.634 (1.35–1.99) | $7.54 \times 10^{-7}$ |
| rs2455925 | 15: 48,893,649 | T/C | Intronic | 9.97 | 1.625 (1.34–1.97) | $9.37 \times 10^{-7}$ |
| rs4775769 | 15: 48,939,888 | G/T | Intergenic | 9.56 | 1.644 (1.35–2.01) | $9.70 \times 10^{-7}$ |

Chr:BP denotes the chromosome location and NCBI Build 37 SNP physical position. MAF, minor allele frequency; OR, odds ratio.

and clinical information on 178,899 Finnish cohort, this haplotype was associated with aortic dissection (p-value $2.3 \times 10^{-5}$; S3 Fig). Interestingly, GWAS of thoracic aorta images in the UK Biobank revealed similar association of FBN1 with dilated aorta [30]. Thus, our study strengthens the emerging functional association between *FBN1* and nonsyndromic aortopathy [31].

## Relationship between pulse rate and prevalence of aortic aneurysms

UK Biobank contains a wealth of baseline clinical information of participants, including height, weight, body mass index, blood pressure and pulse rate [11]. Our UKB OASIS (Omics Analysis, Search & Information System) permits high-throughput analysis of associations between clinical and genetic information (unpublished), When we analyzed AA prevalence by baseline characteristics of BMI, height, waist circumference, blood pressure and heart rate (Table 1), an unexpected correlation emerged between baseline heart rate and prevalence of thoracic aortic aneurysms (TAA). For TAA, but not AAA, there was a general trend toward increased prevalence in individuals with bradycardia (defined as heart rate $\leq$54 beats per minute), regardless of genotype (Fig 3B). Interestingly, this trend seems more marked for homozygous carriers of *FBN1* variants (Fig 3C). In contrast, a general trend toward slightly increased AAA prevalence is seen with tachycardia, although the effect is smaller overall (S4 Fig).

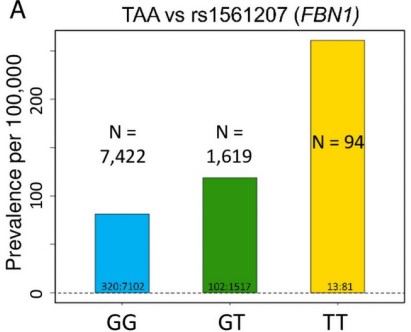 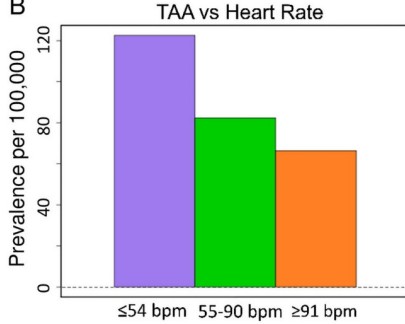 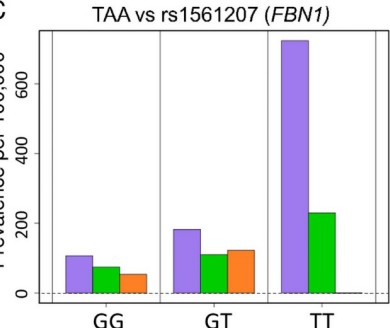

**Fig 3. Prevalence of TAA increases with *FBN1* variant rs1561207 status and bradycardia. (A)** Increase in TAA prevalence per 100,000 participants in the UK Biobank is noted in the homozygotes for the minor allele (T/T) in comparison to the heterozygotes (C/T) and the noncarriers (C/C) in a pronounced stepwise, "dosage-dependent" manner (P-value = $1.57 \times 10^{-6}$, OR per T allele 1.615). Bars labeled with ratio of cases: Controls. **(B)** Prevalence of TAA rises with bradycardia (purple, heart rate $\leq$ 54 beats per minute, bpm) in a stepwise manner from tachycardia (orange, defined as heart rate $\geq$ 91 bpm, OR = 1.89) to normal rate (green, heart rate 55 to 90 bpm, OR = 1.62) to bradycardia (purple, heart rate $\leq$ 54 bpm, OR = 2.09). P-value = 0.01622 by Pearson's Chi-squared test. **(C)** This relationship is seen in both *FBN1* variant carriers (GT) and noncarriers (GG), but the impact of bradycardia is more dramatic in homozygous variant carriers (TT).

## Discussion

Genome wide association analysis of abdominal and thoracic aortic aneurysmal disease in the UK Biobank revealed novel loci near *LINC01021*, *ATOH8* and *JAK2* genes associated with AAA, and novel loci near *CTNNA3*, *FRMD6* and *MBP* genes associated with TAA. Out of the 24 loci previously established for AAA, three were replicated by our analysis, *ADAMTS8*, *CELSR2* and *CDKN2B-AS1* [15,32]. Based on the data compiled here with the thresholds for p-value and minor allele frequency as set forth in the Methods section, there was no significant overlap in the SNPs associated with AAAs and those associated with TAAs. While this is consistent with growing evidence for a distinct underlying genetic architecture and a distinct pathophysiology of these two aortopathies [33], further studies are necessary to specifically address this question.

A potentially clinically relevant finding is the identification of a linkage group of SNPs encompassing the *FBN1* gene, which is associated with TAA in the UK Biobank and with aortic dissection in FinnGen cohorts. This haplotype demonstrated a pronounced dose-dependence, with homozygous carriers associated with ~2.2-fold higher prevalence of thoracic aortic aneurysm than heterozygotes. Given the relatively high minor allele frequencies for these SNPs (9.56–9.97%), as well as the well-defined role of *FBN1* in the pathogenesis of connective tissues disorders including Marfan syndrome, we hypothesize that mutations within this linkage group may account for a non-trivial portion of nonsyndromic thoracic aortic aneurysms and dissections, particularly those within the context of positive family history. Therefore, these variants could merit inclusion in genetic screening panels for familial thoracic aneurysmal disease. Taken together with the findings of Pirruccello, et al., our finding suggests some degree of shared pathophysiology between aortic disease in Marfan syndrome and sporadic thoracic aortic aneurysm [30].

An unexpected finding of this study is the apparent association of TAA prevalence and baseline bradycardia. It is unknown whether bradycardia is a consequence of beta-adrenergic blocker usage in those diagnosed with TAA. Indeed, higher percentage of AAA cases are on beta-blockers than controls (24.3% versus 10.3%; S7 Table). Nonetheless, the fact that this association is not seen with AAA even though AAA cases are also prescribed beta-blockers at a higher rate than their respective controls (24.1% versus 12.8%; S7 Table), suggests some biological basis and warrants further investigation, particularly for those with *FBN* variants.

The approach used in this paper has several limitations. As with any GWAS study, the discovery of novel loci associated with aortopathies does not prove functional causality, and the findings described herein need to be validated by analysis of other databases, ideally in a patient population of more diverse genetic origins than the UK Biobank. There are certain limitations inherent to a population study based on ICD10 codes in comparison to a study dedicated specifically to aortopathies. For example, an ICD10-based studies are limited by the fact that, as in many real-world situations, many diseases and medical conditions are underdiagnosed. In a UKB-based study, this is especially important with respect to the matched controls since they are selected randomly from a pool of individuals who simply do not carry the ICD10 codes, rather than those specifically ruled out for the disease by a focused survey. For instance, valvular disorders are a common co-morbidity of aortic aneurysm, and aortic dissection is a complication of aortic aneurysm (S1 Table), but these subjects may not be coded as having aortic aneurysm per se and inappropriately included among the controls. To account for this, we excluded from controls not just the subjects with ICD10 codes corresponding to aortic aneurysms, but also those with common comorbid conditions and complications of aortic aneurysms, to increase the probability that the controls are truly free of the diagnosis we are

studying. We acknowledge this may have introduced small bias for detecting genetic associations indirectly related to aortopathies.

We also note that our study identified fewer associations than recent GWAS studies on the Million Veterans Program (MVP) cohort and on aortic images in the UK Biobank [15,30]. The fewer associations we identified compared to the MVP study may due to the fact that we examined 1,363 patients with AAA whereas the MVP study examined 7,642 patients [15]. Additionally, the greater number of associations found by Pirruccello et al. could reflect the fact that they used imaging to identify individuals with subclinical aortopathy not captured by ICD10 codes [30]. Finally, we also note that, in addition to the variants and loci discussed here, there are many more that didn't make the genome-wide significance cutoff of $p < 5$ x $10^{-8}$ or MAF cutoff $\geq 0.5\%$. Thus, much of the genetic underpinnings of abdominal and thoracic aortic aneurysm formation remain to be discovered.

## Supporting information

**S1 Fig. Principle components (PCs) by Ethnicity for UK Biobank participants.** When selecting controls for comparison with cases, control subjects were picked from subjects within 80 units on the PC1 vs. PC2 graph. The size of 80 units is illustrated with the red boxes around subjects who are primarily European, Chinese or African Ethnicity based on the PC1 and PC2 eigenvalues provided by the UK Biobank.
(TIF)

**S2 Fig. Quantile-quantile plots (QQ Plots) for the AAA and TAA phenotypes showing that the quality of the association analysis is well controlled with minimal confounding present.** The genomic control (GC Lambda) values of 1.04 (AAA) and 1.05 (TAA) are within the generally accepted range for GWAS.
(TIF)

**S3 Fig. In FinnGen cohort, *FBN1* variant rs625034 is associated with increased prevalence of aortic dissection (P = 2.3 x 10–5).** Manhattan plot of phenome wide association study (PheWAS) is shown.
(TIF)

**S4 Fig. Association between pulse rate and prevalence of abdominal aortic aneurysm formation.** A general trend toward slightly increased AAA prevalence is seen with tachycardia.
(TIF)

**S1 Table. ICD10 diagnostic codes excluded from controls for GWAS of Abdominal Aortic Aneurysm (AAA).**
(XLSX)

**S2 Table. ICD10 diagnostic codes excluded from controls for GWAS of Thoracic Aortic Aneurysm (TAA).**
(XLSX)

**S3 Table. Comorbidities of patients with one copy of the SNPs described in the linkage group encompassing *FBN1* are similar to comorbidities of all patients with TAA.** This table includes comorbidities with frequency $\geq 25\%$.
(XLSX)

**S4 Table. Age at diagnosis for all TAA patients, and for TAA patients with at least one copy of the SNPs described in the linkage group encompassing *FBN1*.**
(XLSX)

**S5 Table. Complete results for SNPs associated with AAA.**
(XLSX)

**S6 Table. Complete results for SNPs associated with TAA.**
(XLSX)

**S7 Table. Beta-blocker usage in AAA and TAA cases, and in respective controls.**
(XLSX)

## Acknowledgments

This research was conducted using the UK Biobank Resource under Application Number 49852. The funders had no role in study design, data collection and analysis, decision to publish, or preparation of the manuscript.

## Author Contributions

**Conceptualization:** Tamara Ashvetiya, Sherry X. Fan, Charles H. Williams, Jeffery R. O'Connell, James A. Perry, Charles C. Hong.

**Data curation:** Tamara Ashvetiya, Sherry X. Fan, Yi-Ju Chen, Charles H. Williams, James A. Perry, Charles C. Hong.

**Formal analysis:** Tamara Ashvetiya, Sherry X. Fan, Yi-Ju Chen, Charles H. Williams, Jeffery R. O'Connell, James A. Perry, Charles C. Hong.

**Funding acquisition:** Charles C. Hong.

**Investigation:** Tamara Ashvetiya, Sherry X. Fan, Yi-Ju Chen, Charles H. Williams, James A. Perry, Charles C. Hong.

**Methodology:** Tamara Ashvetiya, Sherry X. Fan, Yi-Ju Chen, Charles H. Williams, Jeffery R. O'Connell, James A. Perry, Charles C. Hong.

**Project administration:** James A. Perry, Charles C. Hong.

**Resources:** James A. Perry, Charles C. Hong.

**Software:** Yi-Ju Chen, James A. Perry, Charles C. Hong.

**Supervision:** James A. Perry, Charles C. Hong.

**Validation:** Tamara Ashvetiya, Sherry X. Fan, Yi-Ju Chen, James A. Perry, Charles C. Hong.

**Visualization:** Yi-Ju Chen, James A. Perry, Charles C. Hong.

**Writing – original draft:** Tamara Ashvetiya, Sherry X. Fan, Yi-Ju Chen, Charles H. Williams, James A. Perry, Charles C. Hong.

**Writing – review & editing:** Tamara Ashvetiya, Sherry X. Fan, Jeffery R. O'Connell, James A. Perry, Charles C. Hong.

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
