## [Decision Letter · Decision Letter 0]

7 Apr 2021

PONE-D-21-03668

Identification of Novel Genetic Susceptibility Loci for Thoracic and Abdominal Aortic Aneurysms Via Genome-Wide Association Study Using the UK Biobank Cohort

PLOS ONE

Dear Dr. Hong,

Thank you for submitting your manuscript to PLOS ONE. After careful consideration, we feel that it has merit but does not fully meet PLOS ONE’s publication criteria as it currently stands. Therefore, we invite you to submit a revised version of the manuscript that addresses the points raised during the review process.

The reviewers raised significant concerns about methods, analyses, clarity, and language. Please address each comment carefully, changing the manuscript accordingly. Please notice that all data must be provided as part of the manuscript or its supporting information, or deposited to a public repository. For more information, check PLOS Data policy. 

We look forward to receiving your revised manuscript.

Kind regards,

Danillo G Augusto

Academic Editor

PLOS ONE

Journal Requirements:

2. Please amend either the title on the online submission form (via Edit Submission) or the title in the manuscript so that they are identical.

[This research was conducted using the UK Biobank Resource under Application Number 49852. This work was supported by NIGMS R01GM118557 and NHLBI R01HL1351291 to CCH, and NHLBI 1U01HL137181to JP. The funders had no role in the design and conduct of the study; collection, management, analysis and interpretation of the data; preparation, review or approval of the manuscript; or decision to submit the manuscript for publication.]

 [The funders had no role in study design, data collection and analysis, decision to publish, or preparation of the manuscript.]

4. Please include a copy of Table 5 which you refer to in your text on page 14.

Reviewers' comments:

Reviewer's Responses to Questions

**Comments to the Author**

1. Is the manuscript technically sound, and do the data support the conclusions?

Reviewer #1: Partly

Reviewer #2: Yes

2. Has the statistical analysis been performed appropriately and rigorously? 

Reviewer #1: Yes

Reviewer #2: Yes

3. Have the authors made all data underlying the findings in their manuscript fully available?

Reviewer #1: No

Reviewer #2: Yes

4. Is the manuscript presented in an intelligible fashion and written in standard English?

Reviewer #1: No

Reviewer #2: Yes

5. Review Comments to the Author

Reviewer #1: The authors performed a genome-wide association study (GWAS) on abdominal aortic aneurysms (AAA) and thoracic aortic aneurysms (TAA) using data from the UK Biobank. They identified three new risk loci for AAA and replicated three existing loci. For TAA, they also identified three new risk loci. For both traits, no GWAS has been performed in the UK Biobank, as existing efforts focused on aortic aneurysms as a whole. For AAA, two large GWAS non-overlapping efforts have been published before (Klarin, et al. Circulation. 2020, and Jones, et al. Circ Res. 2017), studying 7,642 and 10,204 cases, respectively. For TAA, a GWAS was performed in 2011 on 765 cases (LeMaire, et al. Nat Genet. 2011). None of these studies included the UK Biobank. Although the current study does not outperform earlier efforts by sheer sample size, the release of GWAS summary statistics from the UK Biobank using a thorough analysis pipeline could have added value for the scientific community.

General remarks:

I have some general concerns about the study

1) The GWAS quality control and methods in general are lacking a lot of essential details making assessment of the quality of the study methods difficult. I am also missing quantification of confounding (lambdaGC, LD-score regression intercept). I provide suggestions in the point-by-point comments.

2) The AAA scientific community would benefit most from a meta-analysis of existing GWAS datasets. I think making summary statistics available is a good first step, but I would like to urge the authors to pursue a joint effort.

3) The authors claim that the lack of overlapping risk loci between AAA and TAA identified in the present study is evidence for distinct disease mechanisms. I think the analyses performed in the study are not sufficient to conclude this. I urge the authors to properly test this hypothesis. In the point-by-point comments below I address where I think the conclusions are incomplete.

Point-by-point comments:

Page 6: “As with any GWAS study, the discovery of novel loci associated with aortopathies does not prove functional causality, and the findings described herein needs to be validated by analysis of other databases, ideally in a patient population of more diverse genetic origins than the UK Biobank.”

The part about replication ideally in a diverse population is missing context. How would this improve the study, or how would this specifically support the abovementioned limitations?

Page 9: The GWAS methods are missing important parameters making thorough assessment of the quality of the methods difficult. In each quality control step, please specify the thresholds used for filtering. Also specify any tools used.

Page 9: Please specify what “recommended genomic analysis exclusions” are.

Page 9: “Subjects with these ICD10 codes were removed from the population of controls to avoid introducing confounding factors, specifically the TAA and aortic valvular disorders, in the analysis.“

I don’t agree with this statement (or I don’t fully understand). If only the controls are filtered to exclude persons with these disorders, you are depleting the controls of these conditions. Thereby introducing bias because the prevalence of these conditions will be higher in cases. If cases with these conditions were also excluded then indeed are you reducing potential confounding. Please clarify and if needed adjust the inclusion criteria.

Page 9: Regarding tolerances. Please provide a table with baseline characteristics of cases and controls separately.

Page 10: The PC-based control selection in not completely clear. I would not be able to reproduce this method. What is a PC unit? This does not seem to be value in an eigenvector as +400 is rather large. Please also provide a supplementary figure showing the first few PCs plotted (perhaps against hapmap samples) highlighting the cases and controls in order to see the overlap and distance.

Page 10: “Our preliminary analysis showed that only the first 5 PCs had significance.” What does this mean and how was this tested?

Page 10: “The potential functional significance of associated variants was assessed by Eigen PC scores, presence of promoter or enhancer elements, and presence of DNase hypersensitivity sites in the affected regions of the genome.”

Does this mean that the variants of interest are present in promoter or enhancer elements, or that these are nearby (and within what window)? Provide a reference for Eigen-PC describe the method for obtaining these.

Page 11: The authors do not provide any estimate for presence of confounding. Please provide a lambdaGC value for both GWAS and preferable also an LD-score regression intercept or equivalent metric.

Page 11: “Genetic ancestry was predominantly British (93.54%); however, patients were also represented from Irish (2.64%), Indian (0.22%), Caribbean (0.51%), and African (0.15%) backgrounds“.

Mixed ancestry could bias the association analyses if not accounted for properly. It is the major source for confounding in GWAS in general. Indeed, the authors perform a PC-matched analysis and includes PCs as covariates, but this does not guarantee to fully avoid confounding. It is essential to quantify confounding (as mentioned in an earlier comment). I would suggest to perform a sensitivity analysis excluding non-European ancestry persons and use the metrics for confounding, as well as the Manhattan plots, as comparison.

Page 11 and Figure 1: some loci (especially AAA chromosome 2) contain lonely associated SNPs without accompanying LD SNP. This is not necessarily bad, but it would be good to see some additional metrics: 1) Hardy-Weinberg equilibrium P-values added to supplementary Tables 5 and 6, 2) sensitivity analysis excluding non-European ancestry samples, and 3) are there nearby SNPs that just fall below the MAF threshold and not plotted for that reason?

Page 11. Typo: “(p-value <1e-6; Table 2)”

Page 11: “In addition, we found several distinct variants that do not reach the definitive threshold for genome-wide significance but nevertheless possess a strong basis for biologic plausibility and are within the suggestive threshold for genome-wide significance (p-value <1e-6; Table 2).”

In my opinion it is too much to use the Eigen-PC score to compensate for lack of statistical association. In any genomic region there will be some SNPs that are functionally relevant, but this does not imply disease specificity. The score should rather be used to prioritize SNP within a risk locus. If suggestive SNPs were also found at a suggestive level of significance in other GWAS of AAA/TAA, or are bona fide loci in related traits, that would be a good motivation to prioritize these, rather than by Eigen-PC score alone.

Page 12: “The significant linkage group of ADAMTS8 variants that we identified includes rs7936928 (intronic), rs4936099 (intronic), rs11222084 (intergenic), and rs3740888 (intronic); these variants have p-values 7.51 x 10-9-1.59 x 10-8, MAF 36.7%-40.7%, and odds ratios 0.785-0.790”.

These four SNPs are described as a linkage group, meaning (if my interpretation is correct) that these are in strong LD. This means their P-values (and MAFs) are closely related. Mentioning all four of them could lead to an unwanted feeling of replication or importance to the reader. Reporting only the lead SNP or some prioritized SNP would be sufficient.

Page 12: Top SNPs are linked to genes. There is no clear description of the motivation of prioritizing genes. It looks like the nearest genes are selected, or some nearby gene with disease relevance. I would like to see a data-driven approach to gene prioritization, for which there are many different options.

Page 12: “In addition, we identified several distinct variants that do not reach the standard threshold for genome-wide significance for association with AAA, but are nevertheless within the suggestive threshold for genome-wide significance (p-value <1 x 10-6) and possess a strong basis for biologic plausibility (Table 2).”

Indeed, finding these variants which are bona fide players in other diseases is important. However, the term biological plausibility in this context is ambiguous. The finding supports a pleiotropic role for these loci and indeed support that “something is happening” at that locus. The term biological plausibility/importance implies some functional evidence.

Page 13: “Of these variants, rs12740374, rs660240 and rs7528419 have a particularly high Eigen PC score of 4.4, 3.8 and 3.2, respectively, suggesting a functional role (Table 2).”

Please provide context of the interpretation of specific value of the Eigen score.

Page 13: The same remark about ancestry as for AAA applies here.

Page 14. Typo: “In addition, we identified we identified a linkage group of high-frequency variants”

Page 14: “but nevertheless have a strong basis for biologic plausibility since they fall in a large region of linkage disequilibrium encompassing FBN1 (Table 5).”

Here, the term biologic plausibility is again confusing. Indeed, the position of the hit near FBN1 is interesting, but is no real genetic or biological evidence for these variants. Are there specific SNPs or LD-SNPs identified in earlier GWAS of TAA or related traits?

Page 14: “(Figure 3A; p-value 1.57 x10-6, MAF 9.89%, OR 1.615) is of special interest given its high Eigen PC score”.

Please provide Eigen-PC score in the text.

Page 14: “Interestingly, each of these FBN1 variants demonstrated a pronounced dose- dependence: homozygotes had significantly higher prevalence of thoracic aortic aneurysm than heterozygotes (Figure 3A)”. Note the use of “significantly”.

This implies the difference between heterozygotes and homozygous carriers was tested statistically. Also note that “each of these variants” does not add information over “the variants / the lead variant”, but could lead to an unwanted sense of replication, while these variants are highly correlated.

Page 15: It is unclear why the authors studied the effect of pulse rate on TAA and AAA. Please clarify what the hypothesis was, or whether this was a coincidental finding.

Page 15 and Figure 3B: When we analyzed AA prevalence by baseline pulse rate for each of the statistically significant SNPs identified in this study”.

This effect seems present at some extent when looking at the plots. However, the authors do not provide a statistical test for this effect. To claim this effect to be true, it is essential to properly test this. There are probably several ways, but one would be to add an interaction term in the PLINK model.

Page 16: “Out of the 24 loci previously established for AAA, three were replicated by our analysis, ADAMTS8, CELSR2 and CDKN2B-AS1(14, 31).”

Please describe what the authors define as being replicated (for example: they were genome-wide significant in the present study). I did not see a specific analysis looking at overlap of loci.

Page 16: “Based on the data compiled here with the thresholds for p- value and minor allele frequency as set forth in the methods section, there was no significant overlap in the SNPs associated with AAAs and those associated with TAAs. This suggests a distinct underlying genetic architecture, and a distinct pathophysiology, of these two aortopathies.”

I don’t think the (lack of) overlap was thoroughly tested. I agree that the specific loci identified given the power of the studies do not overlap, but this is not in any way sufficient to conclude that the diseases have distinct genetic architecture or pathophysiology. At least, I would like to see a genome-wide comparison of the traits, for example by genetic correlation analysis using LD-score regression. If power is too low for genetic correlation, a genome-wide replication could be done. Please also include the other GWASs of AAA and TAA in these analyses to make sure lack of overlap is not due to lack of power alone.

Page 16: “It is unknown whether bradycardia is a consequence of beta-adrenergic blocker usage in those diagnosed with TAA.”

The UK Biobank has data on medication use. I don’t know how detailed this is for beta-blockers, but a sensitivity analysis excluding persons using this medication could see if this effect holds and be a potential biologically relevant effect.

Page 17: “ideally in a patient population of more diverse genetic origins than the UK Biobank.”

I would like to see some context on how this would improve our understanding, prevention or treatment. Without context if feels a bit like an empty gesture.

The methods section is missing important details. This makes reviewing the methods difficult.

-I think the English language of the manuscript could be improved. Perhaps a check by a native speaker could be beneficial. The order of word is not always optimal, and some terms are used incorrectly or abundantly

-There are two other GWAS studies of AAA published in the last four years, without overlapping samples and by different groups. I think the scientific community would benefit most from a joint analysis of these datasets. It seems that both other efforts have some sort of restriction on the use of their data. I don't know if this has any historical reasons, but in my opinion it would be good to encourage the authors to pursue collaboration.

-I was unable to find a data availability statement for the GWAS summary statistics (including all SNP effect, and not just the significant ones as provided in Supplementary Tables 5 and 6).

Reviewer #2: Very nicely conducted study. Important. Thank you.

I agree that the bradycardia finding is likely a "red herring", probably reflecting medication difference between the groups.

Where do you suggest that the work in this area go from here? How should your findings be reflected in further scientific and clinical work?

6. PLOS authors have the option to publish the peer review history of their article (what does this mean?). If published, this will include your full peer review and any attached files.

Reviewer #1: **Yes: **Mark K. Bakker

Reviewer #2: **Yes: **John A. Elefteriades, MD

---

## [Author Response · Author response to Decision Letter 0]

8 Jun 2021

Responses to Reviewer Comments. Our responses are Italicized.

 1) Reviewer #1: The authors performed a genome-wide association study (GWAS) on abdominal aortic aneurysms (AAA) and thoracic aortic aneurysms (TAA) using data from the UK Biobank. They identified three new risk loci for AAA and replicated three existing loci. For TAA, they also identified three new risk loci. For both traits, no GWAS has been performed in the UK Biobank, as existing efforts focused on aortic aneurysms as a whole. For AAA, two large GWAS non-overlapping efforts have been published before (Klarin, et al. Circulation. 2020, and Jones, et al. Circ Res. 2017), studying 7,642 and 10,204 cases, respectively. For TAA, a GWAS was performed in 2011 on 765 cases (LeMaire, et al. Nat Genet. 2011). None of these studies included the UK Biobank. Although the current study does not outperform earlier efforts by sheer sample size, the release of GWAS summary statistics from the UK Biobank using a thorough analysis pipeline could have added value for the scientific community.

We agree with the reviewer in principle; however, If we were to provide all summary statistics for the full GWAS, it would be this data for 40 million variants for each trait, for a total of 80 million sets of summary statistics. Presently, UK Biobank has stated that it are not ready to receive GWAS summary statistics, but it will be providing a mechanism for returning GWAS results for sharing with others. When that mechanism is in place, we will be sending our GWAS summary statistics. In meanwhile, we have provided summary statistics for variants with MAF > 0.5% and p-value < 1E-6 in Supplementary Table 5 & 6. 

I have some general concerns about the study.

2) The GWAS quality control and methods in general are lacking a lot of essential details making assessment of the quality of the study methods difficult. I am also missing quantification of confounding (lambdaGC, LD-score regression intercept). I provide suggestions in the point-by-point comments.

These are provided in Supplementary Figures 2 (see also #11, #12).

3) The AAA scientific community would benefit most from a meta-analysis of existing GWAS datasets. I think making summary statistics available is a good first step, but I would like to urge the authors to pursue a joint effort.

The nature of MTA precludes us from directly sharing data outside our home institution. While efforts are in place at UK Biobank to eventually allow sharing of data with the greater research community, such joint effort is beyond the scope of this work.

The authors claim that the lack of overlapping risk loci between AAA and TAA identified in the present study is evidence for distinct disease mechanisms. I think the analyses performed in the study are not sufficient to conclude this. I urge the authors to properly test this hypothesis. In the point-by-point comments below I address where I think the conclusions are incomplete.

Point-by-point comments:

4) Page 6 and 15: “As with any GWAS study, the discovery of novel loci associated with aortopathies does not prove functional causality, and the findings described herein needs to be validated by analysis of other databases, ideally in a patient population of more diverse genetic origins than the UK Biobank.”

The part about replication ideally in a diverse population is missing context. How would this improve the study, or how would this specifically support the above-mentioned limitations?

This is general statement of the limitations. As the reviewer states, meta-analysis of existing GWAS datasets would support this statement; however, this is beyond the scope of the present story.

5) Page 9: The GWAS methods are missing important parameters making thorough assessment of the quality of the methods difficult. In each quality control step, please specify the thresholds used for filtering. Also specify any tools used.

We have added information for UKB Data-Field’s for each item used in the quality control step. If it would be better to use references/links, they are:

https://biobank.ctsu.ox.ac.uk/crystal/field.cgi?id=22018

https://biobank.ctsu.ox.ac.uk/crystal/field.cgi?id=22019

https://biobank.ctsu.ox.ac.uk/crystal/field.cgi?id=31 and

https://biobank.ctsu.ox.ac.uk/crystal/field.cgi?id=22001

https://biobank.ctsu.ox.ac.uk/crystal/field.cgi?id=22010

https://biobank.ctsu.ox.ac.uk/crystal/field.cgi?id=22077

6) Page 9: Please specify what “recommended genomic analysis exclusions” are.

We have added UKB Data Field info, as above.

7) Page 9: “Subjects with these ICD10 codes were removed from the population of controls to avoid introducing confounding factors, specifically the TAA and aortic valvular disorders, in the analysis.“

I don’t agree with this statement (or I don’t fully understand). If only the controls are filtered to exclude persons with these disorders, you are depleting the controls of these conditions. Thereby introducing bias because the prevalence of these conditions will be higher in cases. If cases with these conditions were also excluded then indeed are you reducing potential confounding. Please clarify and if needed adjust the inclusion criteria.

We believe our approach attempts to overcome certain inherent limitations of a population study based on ICD10 codes in comparison to a study dedicated specifically to aortopathies. An ICD10 based studies are limited by the fact that, as in many real-world situations, many diseases and medical conditions are underdiagnosed. This is especially important the controls in a UKB-based study since they are selected randomly from a pool of those who simply do not carry the ICD10 codes, and not specifically ruled out for the disease by a focused survey. For instance, valvular disorders are common co-morbidity of aortic aneurysm, and aortic dissection is a complication of aortic aneurysm, but those subjects may not be coded as aortic aneurysm per se and inappropriately included among the controls. Thus, for controls, we purposely excluded subjects with ICD10 codes but also common comorbid conditions and complications of aortic aneurysms, to increase the probability that the controls are truly free of the diagnosis we are studying. The UK Biobank is a very large cohort so there is ample supply of controls. We have included this explanation in the discussion.

8) Page 9: Regarding tolerances. Please provide a table with baseline characteristics of cases and controls separately.

These have been included in Table 1.

9) Page 10: The PC-based control selection in not completely clear. I would not be able to reproduce this method. What is a PC unit? This does not seem to be value in an eigenvector as +400 is rather large. Please also provide a supplementary figure showing the first few PCs plotted (perhaps against hapmap samples) highlighting the cases and controls in order to see the overlap and distance.

We have updated the description in the method such that others could easily reproduce what was done. Additionally, we have added Supplementary Figure 2 to show the range that was used for ancestry matching. This addresses the issue and graphically illustrates what was done. A plot as described by the reviewer #2, with all cases and controls would not give a meaningful picture because the 20 cases for each control would obliterate the view and the overlaps would not be discernable.

9) Page 10: “Our preliminary analysis showed that only the first 5 PCs had significance.” What does this mean and how was this tested?

We have updated the manuscript.

10) Page 10: “The potential functional significance of associated variants was assessed by Eigen PC scores, presence of promoter or enhancer elements, and presence of DNase hypersensitivity sites in the affected regions of the genome.”

Does this mean that the variants of interest are present in promoter or enhancer elements, or that these are nearby (and within what window)? Provide a reference for Eigen-PC describe the method for obtaining these.

Given the degree of objections to the sue of EigenPC score, we removed their mention form the manuscript. In any case, the reference for EigenPC Score is as follows: Lonita-Laza, Iuliana, et al. "A spectral approach integrating functional genomic annotations for coding and noncoding variants." Nature Genetics 48.2 (2016): 214.

11) Page 11: The authors do not provide any estimate for presence of confounding. Please provide a lambdaGC value for both GWAS and preferable also an LD-score regression intercept or equivalent metric.

These are provided in Supplementary Figure 2 (see also #2, #12).

12) Page 11: “Genetic ancestry was predominantly British (93.54%); however, patients were also represented from Irish (2.64%), Indian (0.22%), Caribbean (0.51%), and African (0.15%) backgrounds“.

Mixed ancestry could bias the association analyses if not accounted for properly. It is the major source for confounding in GWAS in general. Indeed, the authors perform a PC-matched analysis and includes PCs as covariates, but this does not guarantee to fully avoid confounding. It is essential to quantify confounding (as mentioned in an earlier comment). I would suggest to perform a sensitivity analysis excluding non-European ancestry persons and use the metrics for confounding, as well as the Manhattan plots, as comparison. 

We are providing the QQ Plots (Supp Figure 2), which have the GC Lambda value that show confounding is under control.

Page 11 and Figure 1: some loci (especially AAA chromosome 2) contain lonely associated SNPs without accompanying LD SNP. This is not necessarily bad, but it would be good to see some additional metrics: 

13) Hardy-Weinberg equilibrium P-values added to supplementary Tables 5 and 6; 

Hardy-Weinberg equilibrium are not meaningful here because we have mixed ancestry which, a priori, violates the assumptions that HWE requires.

14) sensitivity analysis excluding non-European ancestry samples, and 

This analysis was done with the UK Biobank data which is 90% European ancestry. Thus, excluding those subjects is not a practical approach because it would reduce the sample size 10-fold.

15) are there nearby SNPs that just fall below the MAF threshold and not plotted for that reason?

We are reporting variants with MAF > 1%. The Manhattan Plot shows variants with MAF > 0.5%, so we already are showing variants “below the filter” (Figure; Supplementary Tables 5 and 6).

16) Page 11. Typo: “(p-value <1e-6; Table 2)” 

Corrected!

17) Page 11: “In addition, we found several distinct variants that do not reach the definitive threshold for genome-wide significance but nevertheless possess a strong basis for biologic plausibility and are within the suggestive threshold for genome-wide significance (p-value <1e-6; Table 2).”

In my opinion it is too much to use the Eigen-PC score to compensate for lack of statistical association. In any genomic region there will be some SNPs that are functionally relevant, but this does not imply disease specificity. The score should rather be used to prioritize SNP within a risk locus. If suggestive SNPs were also found at a suggestive level of significance in other GWAS of AAA/TAA, or are bona fide loci in related traits, that would be a good motivation to prioritize these, rather than by Eigen-PC score alone.

What the reviewer #2 describes is exactly what we did. The Eigen-PC score was used to prioritize variants within a locus and that were already suggestive; however, given the objections, we have removed EigenPC score from the analysis and results.

18) Page 12: “The significant linkage group of ADAMTS8 variants that we identified includes rs7936928 (intronic), rs4936099 (intronic), rs11222084 (intergenic), and rs3740888 (intronic); these variants have p-values 7.51 x 10-9-1.59 x 10-8, MAF 36.7%-40.7%, and odds ratios 0.785-0.790”.

These four SNPs are described as a linkage group, meaning (if my interpretation is correct) that these are in strong LD. This means their P-values (and MAFs) are closely related. Mentioning all four of them could lead to an unwanted feeling of replication or importance to the reader. Reporting only the lead SNP or some prioritized SNP would be sufficient.

We have modified this section accordingly. 

19) Page 12: Top SNPs are linked to genes. There is no clear description of the motivation of prioritizing genes. It looks like the nearest genes are selected, or some nearby gene with disease relevance. I would like to see a data-driven approach to gene prioritization, for which there are many different options.

The tables are labeled as “Nearest Gene” (see column headers for the tables listing genes).

20) Page 12: “In addition, we identified several distinct variants that do not reach the standard threshold for genome-wide significance for association with AAA, but are nevertheless within the suggestive threshold for genome-wide significance (p-value <1 x 10-6) and possess a strong basis for biologic plausibility (Table 2).”

Indeed, finding these variants which are bona fide players in other diseases is important. However, the term biological plausibility in this context is ambiguous. The finding supports a pleiotropic role for these loci and indeed support that “something is happening” at that locus. The term biological plausibility/importance implies some functional evidence.

We have removed “and possess a strong basis for biologic plausibility” 

21) Page 13: “Of these variants, rs12740374, rs660240 and rs7528419 have a particularly high Eigen PC score of 4.4, 3.8 and 3.2, respectively, suggesting a functional role (Table 2).”

Please provide context of the interpretation of specific value of the Eigen score.

This section has been removed. 

22) Page 13: The same remark about ancestry as for AAA applies here.

Please see #12, 13, 14 above.

23) Page 14. Typo: “In addition, we identified we identified a linkage group of high-frequency variants”

Corrected!

24) Page 14: “but nevertheless have a strong basis for biologic plausibility since they fall in a large region of linkage disequilibrium encompassing FBN1 (Table 5).”

Here, the term biologic plausibility is again confusing. Indeed, the position of the hit near FBN1 is interesting, but is no real genetic or biological evidence for these variants. Are there specific SNPs or LD-SNPs identified in earlier GWAS of TAA or related traits?

Mention of biological plausibility has been removed. 

25) Page 14: “(Figure 3A; p-value 1.57 x10-6, MAF 9.89%, OR 1.615) is of special interest given its high Eigen PC score”.

Please provide Eigen-PC score in the text.

Again, we have removed this section. No more Eigen-PC score. 

26) Page 14: “Interestingly, each of these FBN1 variants demonstrated a pronounced dose- dependence: homozygotes had significantly higher prevalence of thoracic aortic aneurysm than heterozygotes (Figure 3A)”. Note the use of “significantly”.

This implies the difference between heterozygotes and homozygous carriers was tested statistically. Also note that “each of these variants” does not add information over “the variants / the lead variant”, but could lead to an unwanted sense of replication, while these variants are highly correlated.

“Each of these variants” have been replaced with “this haplotype.”

27) Page 15: It is unclear why the authors studied the effect of pulse rate on TAA and AAA. Please clarify what the hypothesis was, or whether this was a coincidental finding.

We have added the following. “UK Biobank contains a wealth of baseline clinical information of participants, including height, weight, body mass index, bone mineral density, basic laboratory values, blood pressure and pulse rate. Our UKB OASIS (Omics Analysis, Search & Information System) permits high-throughput analysis of associations between clinical and genetic information (unpublished), When we analyzed AA prevalence by baseline characteristics (Supplementary Table 7) for each of the statistically significant SNPs identified in this study, an unexpected correlation emerged between baseline heart rate and prevalence of aortic aneurysms.

28) Page 15 and Figure 3B: When we analyzed AA prevalence by baseline pulse rate for each of the statistically significant SNPs identified in this study”.

This effect seems present at some extent when looking at the plots. However, the authors do not provide a statistical test for this effect. To claim this effect to be true, it is essential to properly test this. There are probably several ways, but one would be to add an interaction term in the PLINK model.

P-value for nongenetic analysis of pulse rate-TAA association is provided (Figure 3) 

29) Page 16: “Out of the 24 loci previously established for AAA, three were replicated by our analysis, ADAMTS8, CELSR2 and CDKN2B-AS1(14, 31).”

Please describe what the authors define as being replicated (for example: they were genome-wide significant in the present study). I did not see a specific analysis looking at overlap of loci.

 “Replicated” means that we got the same results as other studies identified by the reference numbers 14 and 31 in our statement. We did a literature search to determine this. No specific analysis is necessary to confirm that the genes listed in other articles matches the gene names found in our work. 

30) Page 16: “Based on the data compiled here with the thresholds for p- value and minor allele frequency as set forth in the methods section, there was no significant overlap in the SNPs associated with AAAs and those associated with TAAs. This suggests a distinct underlying genetic architecture, and a distinct pathophysiology, of these two aortopathies.”

I don’t think the (lack of) overlap was thoroughly tested. I agree that the specific loci identified given the power of the studies do not overlap, but this is not in any way sufficient to conclude that the diseases have distinct genetic architecture or pathophysiology. At least, I would like to see a genome-wide comparison of the traits, for example by genetic correlation analysis using LD-score regression. If power is too low for genetic correlation, a genome-wide replication could be done. Please also include the other GWASs of AAA and TAA in these analyses to make sure lack of overlap is not due to lack of power alone.

We have modified the text to reflect the fact that, while these finding are suggestive, further studies are needed.

31) Page 16: “It is unknown whether bradycardia is a consequence of beta-adrenergic blocker usage in those diagnosed with TAA.”

The UK Biobank has data on medication use. I don’t know how detailed this is for beta-blockers, but a sensitivity analysis excluding persons using this medication could see if this effect holds and be a potential biologically relevant effect.

We now include evidence that despite the fact that higher percentages of both TAA and AAA cases were on beta-blockers compared to respective controls, the association with bradycardia is noted only for the TAA, suggesting potential biological basis.

32) Page 17: “ideally in a patient population of more diverse genetic origins than the UK Biobank.”

I would like to see some context on how this would improve our understanding, prevention or treatment. Without context if feels a bit like an empty gesture.

The methods section is missing important details. This makes reviewing the methods difficult.

All the requests for more details in the methods have now been addressed.

33) I think the English language of the manuscript could be improved. Perhaps a check by a native speaker could be beneficial. The order of word is not always optimal, and some terms are used incorrectly or abundantly

Just for information the first and the corresponding authors were born in the USA. English is our native language. In over 150 publications by the corresponding authors, this is the first time this issue has been pointed out to us.

34) -There are two other GWAS studies of AAA published in the last four years, without overlapping samples and by different groups. I think the scientific community would benefit most from a joint analysis of these datasets. It seems that both other efforts have some sort of restriction on the use of their data. I don't know if this has any historical reasons, but in my opinion it would be good to encourage the authors to pursue collaboration.

While this is a wonderful idea, but this is currently not possible since, under the existing MTA/Data Use Agreement UKB data cannot be shared outside our institution. That said, other investigators need to obtain data directly from UKB.

35) -I was unable to find a data availability statement for the GWAS summary statistics (including all SNP effect, and not just the significant ones as provided in Supplementary Tables 5 and 6).

As stated in response to #1, #3 and #34, the UK Biobank will be providing a mechanism for returning GWAS results for sharing with others. When that mechanism is in place, we will be sending our GWAS summary statistics.

Reviewer #2: Very nicely conducted study. Important. Thank you.

Thank you very much.

I agree that the bradycardia finding is likely a "red herring", probably reflecting medication difference between the groups.

We now include evidence that despite the fact that higher percentages of both TAA and AAA cases were on beta-blockers compared to respective controls, the association with bradycardia is noted only for the TAA, suggesting potential biological basis.

Where do you suggest that the work in this area go from here? How should your findings be reflected in further scientific and clinical work?

Our work suggesting a degree of shared pathophysiology between with aortic disease in Marfan syndrome and sporadic thoracic aneurysm, should be followed up. If confirmed, one possible application of this work may be inclusion of the FBN variants reported here in genetic screening panel for familial thoracic aneurysmal disease. Our finding of association of bradycardia with thoracic aortic aneurysm warrant further investigation, particularly for those with FBN variants. These points are raised in the discussions.

---

## [Decision Letter · Decision Letter 1]

12 Aug 2021

Identification of Novel Genetic Susceptibility Loci for Thoracic and Abdominal Aortic Aneurysms Via Genome-Wide Association Study Using the UK Biobank Cohort

PONE-D-21-03668R1

Dear Dr. Hong,

We’re pleased to inform you that your manuscript has been judged scientifically suitable for publication and will be formally accepted for publication once it meets all outstanding technical requirements.

Kind regards,

Danillo G Augusto

Academic Editor

PLOS ONE

Additional Editor Comments (optional):

Reviewers' comments:

Reviewer's Responses to Questions

**Comments to the Author**

1. If the authors have adequately addressed your comments raised in a previous round of review and you feel that this manuscript is now acceptable for publication, you may indicate that here to bypass the “Comments to the Author” section, enter your conflict of interest statement in the “Confidential to Editor” section, and submit your "Accept" recommendation.

Reviewer #1: All comments have been addressed

Reviewer #2: All comments have been addressed

2. Is the manuscript technically sound, and do the data support the conclusions?

Reviewer #1: Yes

Reviewer #2: Yes

3. Has the statistical analysis been performed appropriately and rigorously? 

Reviewer #1: Yes

Reviewer #2: Yes

4. Have the authors made all data underlying the findings in their manuscript fully available?

Reviewer #1: No

Reviewer #2: Yes

5. Is the manuscript presented in an intelligible fashion and written in standard English?

Reviewer #1: Yes

Reviewer #2: Yes

6. Review Comments to the Author

Reviewer #1: The authors provided clear responses to all points addressed. The revised manuscript is clear, technically sounds and presents important findings. I think it is a valuable contribution to science and I support its publication in the journal.

I want to apologise to the authors for my remark regarding the English language. This was a blunt comment and I was wrong to make assumptions about native language. It was a humbling lesson for me and thank you for pointing this out.

Reviewer #2: Thank you for revisions. -------------------------------------------------------------------------------------------------------------------------------------------------------------------------------------------------------------------------------------------------------

7. PLOS authors have the option to publish the peer review history of their article (what does this mean?). If published, this will include your full peer review and any attached files.

Reviewer #1: **Yes: **M. Bakker

Reviewer #2: **Yes: **John A. Elefteriades, MD

---

## [Editor Report · Acceptance letter]

23 Aug 2021

PONE-D-21-03668R1 

Identification of Novel Genetic Susceptibility Loci for Thoracic and Abdominal Aortic Aneurysms Via Genome-Wide Association Study Using the UK Biobank Cohort 

Dear Dr. Hong:

I'm pleased to inform you that your manuscript has been deemed suitable for publication in PLOS ONE. Congratulations! Your manuscript is now with our production department. 

Kind regards, 

on behalf of

Dr. Danillo G Augusto 

Academic Editor

PLOS ONE